# DISTANCE VS. COORDINATE: DISTANCE-BASED EMBEDDING IMPROVES MODEL GENERALIZATION FOR ROUTING PROBLEMS

## ABSTRACT

Routing problems, such as traveling salesman problem (TSP) and vehicle routing problem, are among the most classic research topics in combinatorial optimization and operations research (OR). In recent years, with the rapid development of online service platforms, there has been renewed interest in applying this study to facilitate emerging industrial applications, such as food delivery and logistics services. While OR methods remain the mainstream technique, increasing efforts have been put into exploiting deep learning (DL) models for tackling routing problems. The existing DL methods often consider the embedding of the route point coordinate as a key model input and are capable of delivering competing performance in synthetic or simplified settings. However, it is empirically noted that this line of work appears to lack robustness and generalization ability that are crucial for real-world applications. In this paper, we demonstrate that the coordinate can unexpectedly lead to these problems. There are two factors that make coordinate rather 'poisonous' for DL models: i) the definition of distance between route points is far more complex than what coordinate can depict; ii) the coordinate can hardly be sufficiently 'traversed' by the training data. To circumvent these limitations, we propose to abandon the coordinate and instead use the relative distance for route point embedding. We show in both synthetic TSP and real-world food pickup and delivery route prediction problem that our design can significantly improve model's generalization ability, and deliver competitive or better performance with existing models.

## 1 INTRODUCTION

Inspired by the success of deep models, such as Transformer (Vaswani et al., 2017) in tackling language tasks and graph neural network (GNN) (Scarselli et al., 2008) in dealing with unstructured data, growing number of researchers have been attracted to explore the potential of deep learning (DL) models in dealing with routing problems, a research direction historically being dominated by operations research (OR) methods for decades. Numerous DL models, which have achieved success in other research areas, are applied to solve traditional routing problems, such as traveling salesman problem (TSP) and vehicle routing problem (VRP). More recently, with the urgent requirements from online logistics service platforms, route prediction has also become an emerging research topic. For example, the platform usually needs to predict and evaluate whether a package is 'distance-consuming' if it is dispatched to a courier. The predicted route, as well as related route properties, can be used in these evaluations and is vital for improving platform performance. These two kinds of problems, namely route optimization and route prediction, are also the main focus of this paper.

Routing problems, to a great extent, can be defined by the properties of route points (or called nodes in some literature) and the relationship among them. In light of this, it is not surprising to understand that route point characterization plays an irreplaceable role in the algorithm design. To the best of our knowledge, almost all existing DL models tend to take the route point coordinates or their corresponding embedding as the model input. With such coordinate information, competitive performance are achieved via numerical experiments, mostly conducted on synthetic data. However, when it comes to the real-world data, we empirically note that the coordinate information turns to be 'poisonous', rather than informative. A DL model which employs the coordinate information often

delivers less promising results even after training with large scale dataset. Moreover, by adding noises or perturbations to the coordinate input, the model performance may drop dramatically. In comparison, the classic OR based methods seldom face these problems. This may explain why OR methods are still the de-facto solutions for many industrial-level routing problems - there remains a great practical gap between OR and DL in real-world applications and the generalization ability of DL models still concerns.

We demonstrate that it might be better to abandon the coordinate in order to improve the model generalization ability. More specifically, the lack of generalization ability in many existing DL models is closely related to the 'curse of coordinate'. Two main reasons may support this finding. First, coordinate may not be what we really need for the routing problems. In problems such as TSP, the goal is to minimize the total traveling distance. It suffices to provide the distance between route points, rather than the coordinate information. Moreover, in the real world, the distance information usually relies on a complicated geographic information system. It is far more complex than what simple coordinate can depict.

Figure 1 provides a simple illustration for a TSP. The distance between point A and B becomes much longer when a barrier (e.g., a mountain or a high way) exists, while the distance stays the same through the lens of coordinate. The distance change further results in the optimal solution change. This information mismatch, which is common in real-world data, may significantly decay the model performance. Second, the large-scale data may not be large enough to provide sufficient samples for the real-world coordinate. Apart from inferring the distance from the coordinate,

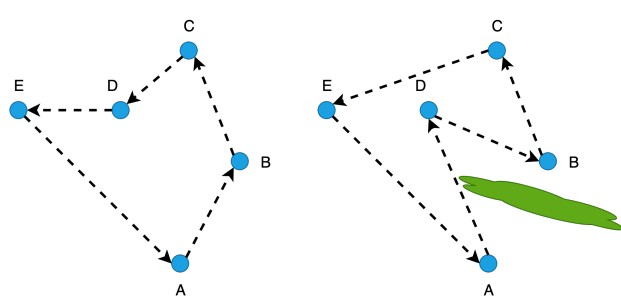

Figure 1: TSP Example: A barrier between point A and B significant changes the optimal solution.

a natural idea can be borrowed from the language tasks. Similarly, we can treat coordinate as word tokens and try to learn the token embedding through large scale data training. However, unlike the coordinate in the synthetic setting, where nearby coordinate are assumed to resemble each other, two nearby coordinates in the real world can be significantly different. Also take Figure 1 as an example, although point A and B with a barrier are close enough in the coordinate space, they should differ drastically due the distance problem. Therefore, the number of possible coordinate explodes in the real world. The DL models may not be sufficiently trained and thus lack generalization ability when using the coordinate.

Our treatment to the 'curse of coordinate' is simple. We propose to use the relative distance between points instead of the coordinate itself. As shown in Figure 2, a common practice in existing deep models is to first project the coordinates into embedding vectors, then feed the embedding vectors into the deep models. We argue that, by simply replacing the coordinate with a distance vector containing distances from the point to all the points (include the point itself), we can achieve significantly better generalization ability and better model performances compared to the existing work. Moreover, our design helps DL models approach to competitive or even better results compared to OR methods in real-world applications. This could pave the way for the large application of DL models in the industry.

The remaining paper is organized as follows. In Section 2, we review the existing DL methods for handling routing problems. In Section 3, we give detailed discussions on why the distance-based embedding outperforms the coordinate-based embedding. In Section 4, we support our insight via experiments conducted on both synthetic and real-world data.

## 2 RELATED WORK

The recent years have witnessed the vibrant use of DL models, such as Transformer (Vaswani et al., 2017) and GNN (Scarselli et al., 2008), for tackling the route optimization and prediction problems.

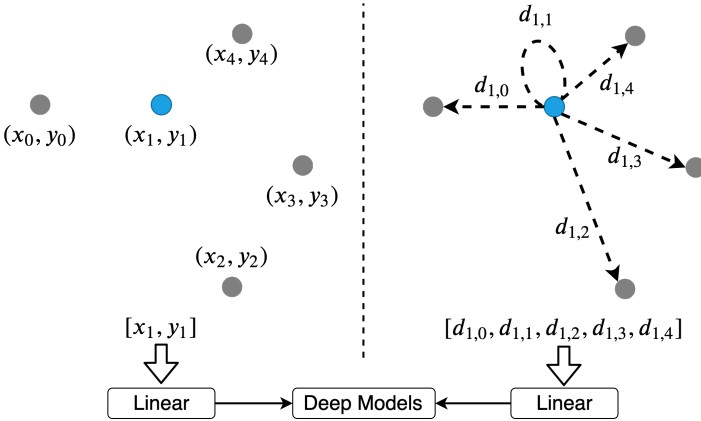

Figure 2: Coordinate-based embedding & distance-based embedding

**Deep Models for Route Optimization.** There are two main kinds of deep model based solutions for route optimization problems, namely construction based methods and improvement based methods. To directly construct optimal solutions, Vinyals et al. (2015) propose the Pointer Network which solves TSP in a supervised learning way. LSTM and attention model are firstly used to achieve competitive results compared to OR methods. Reinforcement learning based methods are then imported to improve the model performance for TSP (Bello et al., 2016), VRP (Nazari et al., 2018) and pickup and delivery problem (PDP) (Ma et al., 2021a). Inspired by the outstanding performance of Transformers in NLP tasks, Kool et al. (2018) provide an attention based model and significantly improve the performance for both TSP and VRP. Compared with RNN based methods, Transformer based models (Kaempfer & Wolf, 2018; Deudon et al., 2018; Ma et al., 2021b; Xin et al., 2021) show great potential in achieving better performances. In order to better mine the spatial relationship among points, GNN based models (Khalil et al., 2017; Nowak et al., 2018; Joshi et al., 2019; Fu et al., 2021) are also explored to generate embedding input or edge probability by aggregating the point and edge features. Different from the construction based methods, improvement based methods (Ma et al., 2021a; Wu et al., 2021; Chen & Tian, 2019; Lu et al., 2019) try to iteratively improve the solution given an initialization. Operations, such as swap and 2-opt, are applied to improve the solution. These model-guided operations are similar to those used in OR areas and can outperform state of the art (SOTA) OR methods in several problem settings. To our knowledge, almost all models aforementioned conduct experiments on the 2D euclidean space or take the point coordinate as the model input.

**Deep Models for Route Prediction.** Route prediction is widely used in logistics and FPD services. The predictions serve as one of the most important inputs for the online service system. Zhang et al. (2019) firstly treat the prediction in package delivery services as a Markov Decision Problem (MDP) and use LightGBM to directly predict which point to serve next. LSTM (Gao et al., 2021), Transformer (Wen et al., 2021; 2022b) and Graph Convolutional Network (GCN) (Wen et al., 2022a) are then used to generate embedding for the candidate serving points. All these models take the point embedding as input and predict the route in a recursive way. Different features are used in these applications, such as coordinate, distance, area ID and other point features. However, we note that a direct comparison between the coordinate and distance is absent.

## 3 METHOD

In this section, we apply a typical routing problem setting and focus only on the route point embedding issue. We will show that it is necessary to re-examine the role of the coordinate information in learning route point embedding, and the distance-based embedding can intuitively perform better and result in better model generalization ability.

### 3.1 PROBLEM FORMULATION

A routing problem instance $r$ typically includes a point set $V = \{v_1, ..., v_n\}$ with $n$ points, a distance matrix $D = \{d_{i,j} \mid i \in [1, n], j \in [1, n]\}$ and extra point features/constraints $X = \{x_i \mid i \in [1, n]\}$, where $v_i$ is the coordinate of the $ith$ point, $d_{i,j}$ is the distance between $v_i$ and $v_j$. A feasible solution for a routing problem can be defined as $\pi = (\pi_1, ..., \pi_n), \pi_k \in [1, n]$, where $\pi_k$ is the index of the $kth$ point in a route and all points in the route must satisfy the constraints. For example, the pickup point should be scheduled before the delivery point in PDP. Two different tasks can be performed under this setting, namely optimization and prediction. For optimization tasks, such as TSP, VRP and route scheduling in PDP, a commonly used optimization goal is to minimize the total traveling distance. For prediction tasks, the goal is to predict the routes which are consistent with real human actions. Even though different optimization or prediction goals lead to various model structure and loss designs, the embedding of the route point is unanimously treated as the essential model input. It has been experimentally shown that a well designed route point embedding can directly benefit DL models for better performance.

### 3.2 DISTANCE VS. COORDINATE

One key information the embedding need to convey is the 'position' of the point. Intuitively, the most explicit representation of a point position might be the coordinate $v_i$'s, with the most popular one being defined on the two-dimensional planar system. Aside from the coordinate, the distance between a point and its neighbors (including the point itself) $d_i = [d_{i,1}, ..., d_{i,n}]$ can also be used to depict the relative point position. Similar to the research in graph, which considers the neighborhood structure as an important property of the node, the distance vector $d_i$ provides a dual representation for the point.

Taking a quick survey on the literature, it is not hard to find that the majority of the work prefers the coordinate to the distance. As a somehow surprising insight, we will argue that the distance may be a way better candidate for routing problems.

**Robustness.** In practice, a robust model is often of main concern, especially for modern online services where an occasional 'bad' case could lead to severe loss. While many OR based methods possess satisfactory robustness, existing DL models which take the coordinate-based embedding as input oftentimes fail to provide robust results. As a typical observation, the DL model performance could drop dramatically even when we simply rotate or translate the point coordinate, as detailed in Section 4.1. Such perturbations do not change the relationship among route points at all, and thus the performance drop is for no reason acceptable. On the contrary, the distance provides us with a more robust representation of the route points, as rotation or translation will not change the relative distance among them. Thanks to this property, consistent solutions can be achieved before and after these perturbations. In addition, it is worthwhile to mention that, suggested by empirical results, models leveraging the distance-based embedding can still provide stable performance even if the relative distance is changed, as long as the distance embedding is well learned during the training. In a nutshell, the distance embedding could endow the DL models with robustness comparable to OR methods.

**Generalization.** It is necessary to study if a model learned on a small dataset can perform well on a larger dataset or if a model learned using one dataset can be used on another dataset. such generalization ability greatly determines the practical potential of the model, particularly in real-world applications where we can hardly ensure sufficient training data on all possible scenarios. Let us take the FPD prediction as an instance, in which a point in a route stands for a restaurant or a recipient in the real world. We can hardly foresee all possible routes that couriers may encounter. However, the sense for distance that couriers hold is arguably similar for different serving regions. As a result, route points that share similar relative distance and constraint properties could be embedded similarly, no matter where they lie in. This is indeed the case for the distance-based embedding, and thus comes naturally the appealing generalization ability for the model using this kind of embedding input. As discussed in Section 4.2, with the help of the distance, a model trained with a smaller dataset collected from one city can provide competitive performance to another model trained with a larger dataset collected nationwide in a FPD route prediction task.

**Intuition.** The route optimization problems can have multiple objectives, such as distance, time and fleet size. Among them, the distance is one of the most important ones. And for route prediction

problems, the distance is also a key factor which impacts the human action. In view of this, it is vital to mine the relationship between the coordinate and the distance when we use them in DL models. To our understanding, when feeding the coordinate to a model, there are two ways to interpret the possible learning process. The first one is, the DL model learns to mimic a 'dictionary' where the search query is the coordinate sequence. In this way, when a similar coordinate sequence appears in the inference, the corresponding record pops up from the dictionary as the prediction. Therefore, the distance information is not really considered in this interpretation. The second interpretation is, the DL model tries to infer the latent distances from coordinates, and then the solution is generated based on the learned distances. This learning process is feasible, since the distance itself is a key objective for many routing problems, e.g., minimizing the total traveling distance in TSP and VRP. So far, there has not been theoretical study on which learning process existing models really take, and this ambiguity makes it challenging to devise tailored learning tasks for this sort of embedding. From a pragmatic point of view, we consider the second interpretation the more reasonable one, which has strong correlation with the learning goal. If so, taking into account that learning distances from coordinates is already a complicated task (as illustrated in Fig. 1), a better way is to leverage the distance embedding, where the model can skip the latent distance learning step and proceed to tackle the final goal directly.

## 3.3 RELATIONSHIP WITH GRAPH BASED EMBEDDING

The graph based models usually take both the node (coordinate) and edge (distance) properties into consideration. For transformer based models, the distance-based embedding can be directly used as a replacement of the coordinate-based embedding. No additional efforts are needed. This simple replacement can benefit the model performance as well as the generalization ability. Experiments of this replacement will be provided in Section 4. For GNN, ideally these models are capable of grasping the neighborhood relationship among points, including the distances. In most existing work using GNN, both the coordinate and the distance are provided as input for the models. It is suggested to remove the coordinate as input regarding the generalization problem. For the distance-based embedding, an identity distance aggregation operator can be designed and acts the same way with the distance vector.

## 4 EXPERIMENTS

In this section, we present two experiments, TSP on synthetic data and FPD route prediction on real-world data. In order to spotlight the comparison between the coordinate and distance based embedding, we do not provide any new models, but use only the existing SOTA TTransformer based DL models and OR methods. Detailed analysis and discussions are provided below.

## 4.1 TRAVELING SALESMAN PROBLEM ON SYNTHETIC DATA

### 4.1.1 SETUP

**Datasets.** The training and test dataset are randomly generated under a coordinate range of $[0, 1]$. To simulate the complicated coordinate distribution in real-world applications, two distributions are adopted in the generation process, i.e., uniform and triangular (the peak is set to $0.3$) distributions. To evaluate the capability of the models for handling complex distance settings, random distances are assigned to point pairs. Specifically, we first generate $10k$ coordinate samples for each distribution, and then compute the distance between two points by multiplying their corresponding Euclidean distance with a random scale drawn from $[1.0, 5.0]$. These 10k points are marked with index range $[0, 10000)$ and their distance matrix are constructed based on the synthetic distances. During the training, route instances are constructed by sampling from the $10k$-point set on the fly. In all trails, we assume the number of customers $N$ to be 20.

**Model & Training.** We use the Attention Model (Kool et al., 2018) in this experiment. Identical to the hyper-parameter setting in the literature, we use a constant learning rate of $\eta = 0.0001$, an epoch number of 100 and 10000 randomly generated route samples for testing. To shorten the running time of each trial, 12.8k training samples are used for each epoch. Comparison are carried out for three types of embedding input, one using the coordinate, one using the distance and the other using both. When both of the embedding are used, they are concatenated and projected to the input dimension

size of the Transformer with a dense layer. Our source code is adapted from the one released in (Kool et al., 2018)[1] and will be published on Github for the repetition of the experiments.

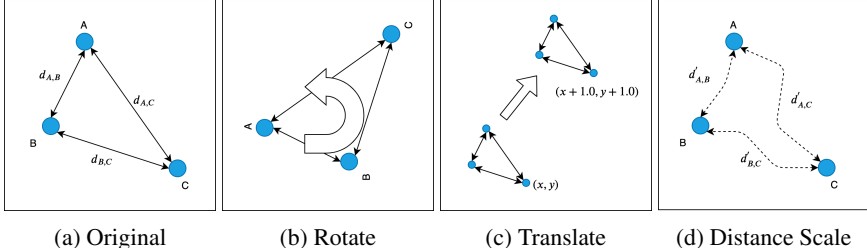

| (a) Original | (b) Rotate | (c) Translate | (d) Distance Scale |

Figure 3: Perturbations: three kinds of perturbations are applied to the coordinate and distance.

**Perturbations.** To verify the robustness and generalization ability of the model, perturbations are applied to the test dataset. Inspired by the robust analysis (Bhojanapalli et al., 2021) for Transformer in computer vision tasks, three perturbation operations are designed: a 90 degree rotation centered on $[0.5, 0.5]$ (named as $Rotate$), a $[1.0, 1.0]$ translation (named as $Translate$) and a random scaling of the distances in the range of $[0.7, 1.3]$ (named as $DistanceScale$). Simple illustrations for these operations are shown in Fig. 3.

**Metric.** The average objective values (the traveling distance that serves all customers) on the test dataset are reported to showcase the performance. The smaller the objective value is, the better performance the embedding achieves.

### 4.1.2 COMPARISON & DISCUSSION

Table 1 summaries the experiment results on the synthetic data. Detailed statistics are provided in Appendix A.1. As for the coordinate-based embedding, since the new coordinates after $Translate$ never appear in the training data, it fails significantly. Interestingly, this embedding shows different (but mild) performance change trends when $Rotate$ is applied in the two distributions, which somehow indicates that the coordinate-based embedding is more sensitive to the rotation under a more complex data distribution. In terms of the distance-based embedding, we can see that it gives the best performance in almost all tasks. Specifically, it delivers completely same results in $Original$, $Rotate$ and $Translate$. This is due to the fact that the relative distance properties among points do not change under these operations. Moreover, when $DistanceScale$ is applied (where part of the distances decrease), it turns out to approach to better solutions. Finally, the situation is more complicated when it comes to the third embedding. It outperforms the coordinate based embedding under the uniformly distributed data, but becomes the worst when the data follow triangular distribution. This echos our previous discussion, where the ambiguity brought by the coordinate information could hamper the learning process.

### 4.2 ROUTE PREDICTION ON FOOD PICKUP AND DELIVERY DATA

The goal of the route prediction is to imitate courier action in food delivery services. Given a set of packages, a courier need to schedule the pickup and delivery point sequences and complete them. The prediction is used in the online service systems to support package dispatch decisions. For example, if the predicted delivery distance of a package is too long for a courier, the package should not be dispatched to him. In this experiment, we compare the performance of different embedding in a real-world FPD data. Compared to the synthetic scenario for TSP, the real-world FPD route prediction problem provides insight for the usage of different embedding in practical applications.

### 4.2.1 SETUP

**Datasets.** The training and test dataset are collected from an online food delivery service system. Two dataset with different sizes are provided in the experiments: a large dataset collected from

---

[1]https://github.com/wouterkool/attention-learn-to-route

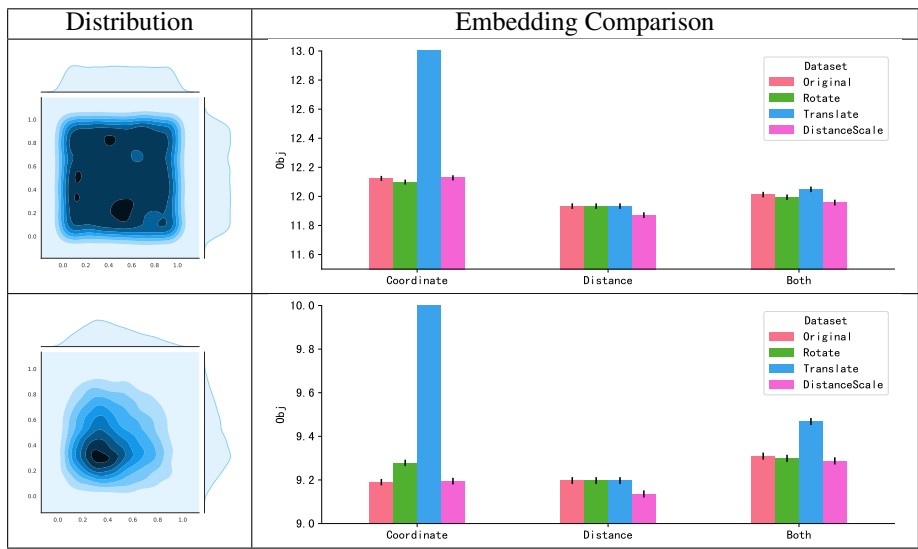

Table 1: Model performance under different data distributions and point embedding.

hundreds of cities (noted as *AllCities*) and a small dataset collected from one specific city (noted as *OneCity*). In which, the large dataset contains 20 million routes for training and $50k$ routes for testing, the small dataset contains 2 million routes for training and also $50k$ routes for testing. All training data are randomly sampled from two weeks ($20220808 \sim 20220821$), and all test data are randomly sampled in the following week ($20220822 \sim 20220828$). Each route include a set of packages and the real pickup and delivery point sequence of a courier. For each package, the position/coordinate of the pickup and delivery point is provided, as well as the navigation distance between each point pair. Additional package features are also provided, such as the estimated time of arrival, the order time and the package status. The package delivery status indicates whether the pickup point of a package have been completed. A package already picked up by the courier only retains a delivery point in the route. One strict constraint needs to be satisfied: the pickup point of a package should be completed before its delivery point. Similar to the evaluation in Wen et al. (2022a), we separate the routes in the system into two parts according to the size of points ($[1, 12]$ and $[12, \sim)$) in a route. Since the processing of long routes is out of the scope of this paper and specific model design is needed to tackle the problem, we only evaluate on the routes with a point size of $[1\ 12]$. Moreover, routes with less than 3 points are also filtered, since routes with a single package already have two points (one pickup and one delivery point) and there is no need to predict such routes. Finally, note that the online dispatch system will continuously dispatch packages to the couriers, a new route is predicted after a new package is dispatched. Therefore, only the route points which are completed during two consecutive dispatch time are evaluated.

**Model & Training.** Following the design of the DL models in (Kool et al., 2018) and (Wen et al., 2021), we use the Transformer for the route point embedding and similarly apply the attention based decoder to generate the predicted route, which is shown in Figure 4. Three main kinds of features are provided as candidate input for the Transformer, namely the point coordinate (longitude and latitude), the point distance and package features related to the point. Different from the model implemented in (Wen et al., 2021), the embedding of the courier's depot point is used as the input to the first decoding step, rather than an initialized one. For the coordinate and the distance, linear embedding are firstly used to project them into a feature vector with a size of 512. DeepFM is applied to the package features. Then all candidate features are concatenated and fed into the Transformer through an additional linear dense layer. The Transformer we use have 3 blocks with 8 heads. The hidden dimension of the Multi-Head Attention (MHA) is 512 and the dimension of the Feed Forward (FF) layer is 2048. We use a batch size of 512, an epoch number of 10 and a learning rate of 0.0001 with exponential decay ($ratio = 0.95$) every 5000 steps in the training. For comparison with existing OR methods, we implement the 2-stage (Zheng et al., 2019) heuristic search algorithm which is specifically designed for the same FPD prediction and already extensively

used to support online services. Simple strategies, such as distance greedy and time greedy, are not provided considering their less competitive performances.

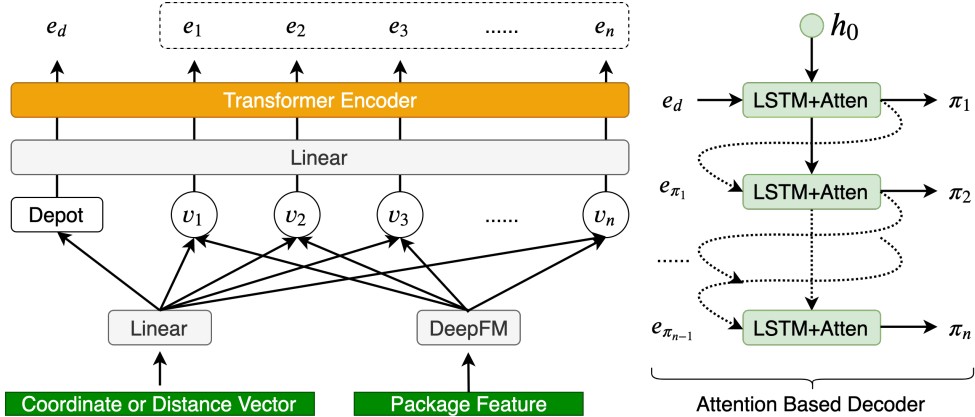

Figure 4: Transformer based model for FPD route prediction.

**Metrics.** The prediction performance is characterized by the similarity between the predicted route $\pi_p$ and the courier's real route $\pi_r$. Formally, given a point $v_i$, the rank of the point in the predicted route and the real courier route is defined as $O_{\pi_p}(v_i)$ and $O_{\pi_r}(v_i)$ correspondingly. Three metrics are used to measure the similarity.

*Kendall Rank Correlation (KRC)(Kendall, 1938):* The KRC is used to measure the relative rank correlation between two sequences. Given a point pair $(v_i, v_j)$, the rank of the point in the predicted route $(O_{\pi_p}(v_i), O_{\pi_p}(v_j))$ and in the courier;'s real route $(O_{\pi_r}(v_i), O_{\pi_r}(v_j))$, the point pair is rank correlated if $(O_{\pi_p}(v_i) - O_{\pi_p}(v_j)) * (O_{\pi_r}(v_i) - O_{\pi_r}(v_j)) > 0$. Otherwise, it is uncorrelated. The KRC is defined as:

$$KRC = (N_c - N_d)/(N_c + N_d)$$

where $N_c$ is the number of correlated point pairs, and $N_d$ is the number of uncorrelated pairs. Greater KRC means higher similarity between two sequences.

*Least Square Deviation (LSD):* The LSD is used to measure the rank deviation of the same point in two sequences.

$$LSD = \frac{1}{n} \sum_{i=1}^{n} (O_{\pi_p}(v_i) - O_{\pi_r}(v_i))^2$$

where $n$ is the number of points in the route. Smaller LSD means higher similarity.

*Consistent Rate (CR):* The CR is used to quantify the ratio of completely same-rank points in two sequences. Given a point $v_i$, if $O_{\pi_p}(v_i) = O_{\pi_r}(v_i)$, the point is reported to be completely same.

$$CR = N_s/N_r$$

where $N_s$ is the number of completely same points, $N_r$ is the number of total points. Greater CR means higher similarity.

### 4.2.2 COMPARISON & DISCUSSION

Table 2 concludes the results of all settings we tried. The model which employs the distance-based embedding and trained on $AllCities$ dataset shows the best result on all the metrics for both test datasets. On the contrary, the model with the coordinate-based embedding performs the worst. We also note that the model trained on $OneCity$ dataset with the distance-based embedding delivers competitive results with the best one. With a smaller size of training data from only one city, the model can provide rather good generalization ability for the data from other cities. A simple change from the coordinate embedding to the distance-based embedding can significantly improves the

model performance on the FPD route prediction task. Finally, while the coordinate-based model still shows worse performance, the distance-based model significantly outperforms the well-designed 2-Stage OR method in all metrics.

| Embedding | Train Data | Test Data | | | | | |
|---|---|---|---|---|---|---|---|
| | | All Cities | | | One City | | |
| | | KRC | LSD | CR | KRC | LSD | CR |
| Coordinate | All Cities | 0.92150 | 0.47843 | 0.78837 | 0.92146 | 0.44971 | 0.78067 |
| | One City | 0.91665 | 0.51529 | 0.77783 | 0.91774 | 0.47549 | 0.77415 |
| Distance | All Cities | **0.94435** | **0.29974** | **0.84037** | **0.93860** | **0.32151** | **0.82410** |
| | One City | 0.93861 | 0.32815 | 0.82656 | 0.93605 | 0.33216 | 0.81758 |
| Both | All Cities | 0.93018 | 0.37834 | 0.80458 | 0.92735 | 0.38471 | 0.79420 |
| | One City | 0.92303 | 0.42768 | 0.78961 | 0.92120 | 0.42445 | 0.78447 |
| 2-Stage | / | 0.93297 | 0.35746 | 0.81080 | 0.92358 | 0.40399 | 0.78882 |

Table 2: Evaluation for different Embedding using different training and test dataset.

### 4.2.3 COORDINATE & DISTANCE-BASED EMBEDDING

In both synthetic and real-world data experiments, the hybrid use of coordinate and distance in embedding shows similar performance trends, i.e. it is not as powerful as the distance-based embedding alone. To shed some light on why this phenomenon occurs, we compare the validation loss of the three models during the training (with *AllCites* dataset in the FPD route prediction experiment). Here, a validation set of size 20k is used. As illustrated in Figure 5, the model with the coordinate-based embedding has the largest loss through the training. Although adding additional distance-based embedding to the model contributes to the loss decrease, it still falls behind the distance-based embedding. One guess is that while the coordinate-based embedding is capable of carrying latent distance information after training with large-scale data, it becomes 'notorious' noise when precise distance is available. Such noise interferes the learning process, resulting in larger validation loss and worse performance.

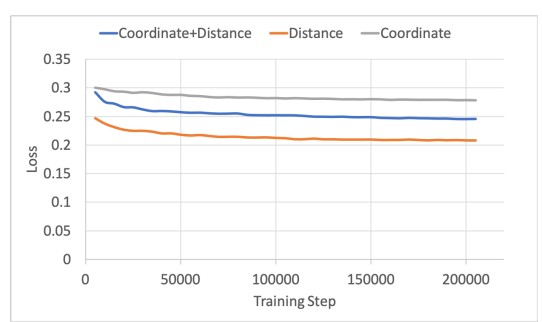

Figure 5: Validation loss using different embedding.

## 5 CONCLUSION

In this work, we study the embedding of route points in DL models for routing problems. In particular, we focus on comparing the popular coordinate-based embedding with the distance-based one. The distance-based embedding outperforms the coordinate-based embedding in both the synthetic TSP and real-world FPD route prediction experiments. Our take home message is as follows: by simply replacing the coordinate-based embedding with the distance-based embedding when handling routing problems, DL models can directly benefit from such replacement and is likely to perform better without any additional efforts.

Finally, from a theoretical point of view, why the model performance drops when using both the coordinate and distance? We would like to end this paper with such a question, as a fundamental research on the correlation between the coordinate and the distance may provide meaningful insight for advancing the research on DL for routing problems.

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

## A    APPENDIX

### A.1    PERFORMANCE STATISTICS FOR THE SYNTHETIC DATA EXPERIMENT

In this section, we provide the detailed statistics for the TSP experiment on the synthetic data, as shown in Table 3 and Table 4.

| Embedding | Original | Rotate | Translate | Distance Scale |
|---|---|---|---|---|
| Coordinate | 12.123 +- 0.017 | 12.098 +- 0.016 | 17.922 +- 0.040 | 12.127 +- 0.018 |
| Distance | 11.933 +- 0.019 | 11.933 +- 0.019 | 11.933 +- 0.019 | 11.870 +- 0.019 |
| Both | 12.012 +- 0.019 | 11.994 +- 0.019 | 12.048 +- 0.019 | 11.957 +- 0.019 |

Table 3: Detailed statistics for the experiment on the uniform sampled data.

| Embedding | Original | Rotate | Translate | Distance Scale |
|---|---|---|---|---|
| Coordinate | 9.189 +- 0.014 | 9.278 +- 0.014 | 11.028 +- 0.018 | 9.194 +- 0.015 |
| Distance | 9.197 +- 0.015 | 9.197 +- 0.015 | 9.197 +- 0.015 | 9.135 +- 0.016 |
| Both | 9.309 +- 0.016 | 9.299 +- 0.016 | 9.467 +- 0.016 | 9.287 +- 0.016 |

Table 4: Detailed statistics for the experiment on the triangular sampled data.

