# OpenReview forum: "Distance VS. Coordinate: Distance Based Embedding Improves Model Generalization for Routing Problems"
_ICLR.cc/2023/Conference — Submitted to ICLR 2023_

### Official Review · Reviewer_jQZV · 2022-10-23

**Confidence:** 4
**Correctness:** 3
**Technical Novelty And Significance:** 1
**Empirical Novelty And Significance:** Not applicable
**Recommendation:** 3

**Clarity, Quality, Novelty And Reproducibility:**

- For the clarifty of the paper, I feel that the paper is well structured and explained. As it is based on the existing well-known learning-based solver (AM), the problem (on embeddings) is well introduced in the first part. The experimental evaluations are well explained and displayed clearly.

- For the quality and originality of the work, I feel that the contribution is a bit weak, particularly for publishing the result in top-tier conferences. The experimental resutls on both synthetic TSP and food delivery data are interesting and meaningful, and I agreed with this that both question and the barrier example in Fig.1 are interesting for routing problems. However, the result on the synthetic data seems to be almost straightforward results. Using distance embedding is also a straightfoward way to embed two dimensional points. If some barrier can be observed, it is natural to embed their information as well (no only points). So, the importance and quality of the the experimental results, at least in the current form, seems to be limited in my opinion.

**Strength And Weaknesses:**

[Strength]
- The paper is well structured and easy to follow the idea and what the authors investigated.
- Numerical evaluations with both real and synthethic data.

[Weakness]
- Technical contributions are not enough to publish on top-tier conferences in my opinion.
- Experimental evaluations are closed within learning-based methods only.

**Summary Of The Paper:**

The paper discusses the embedding part of learning-based routing problems. That is, some existing learning-based solvers tried to embed problem instances by coordinates of points (to be visited by TSP for example), but the paper is investigating whether or not the coordinate-based embedding is poisonous for DL models.

The paper evaluated the concept numericaly based on the existing AM implementation. Using synthethic TSP data (with perturbations by rotations, translations (e.g., Afffine transformation), and distance scaling), the robustness and generalization ability is validated. Using real oline food delivery service system data, with metrics of KRC, LSD, and CR, distance-based embeddings show better results on test data.

**Summary Of The Review:**

The paper studies the embedding part of learning-based routing problems. I agreed with the discussed problem is interesting and the current experimental results on both real and synthetic data are somewhat meaningful, I feel that the current contribution is a bit incremental and then weak to publish the results in top-tier conferences.

---

### Official Review · Reviewer_2rgX · 2022-10-23

**Confidence:** 5
**Correctness:** 4
**Technical Novelty And Significance:** 2
**Empirical Novelty And Significance:** 2
**Recommendation:** 3

**Clarity, Quality, Novelty And Reproducibility:**

There is no particular concern  about reproducibility and clarity, while the novelty is questionable. I find little techncal novelty for addressing the identified problem, which in fact has also been dealt with in existing coordinate-based solvers [2].

**Strength And Weaknesses:**

Strength
- It provides a new perspective for CO researches: improve model robustness for real-world data and real-world perturbations and designing a new distance-based solver is the key to achieve that.

Weakness
- The paper does not provide many insights for improving model robustness when meeting perturbations in real-world cases, e.g. paths are blocked in Fig. 1.
- Some concepts are not clear, e.g. 'posionous' data, 'traversed' in the abstract, 'curse of coordinate'. What do these concepts mean specifically?
- The significance of the approach is questioned: we already have distance-based CO solver [1], what is the significance of feeding distance data to coordinate-based solvers [2]?
- No impressive results are given, especially model performance compared with traditional solvers.

**Summary Of The Paper:**

The paper claims that distance-based CO solvers are more robust than coordinates-based solvers, with concerns for two defects of DL solvers: i) coordinates can hardly depict real-world cases, and ii) coordinates can not be sufficiently 'traversed' by training data.

**Summary Of The Review:**

The paper provides some observation and give some discussion on the robustness of learning based solvers when they are trained using coordinates or distance, and suggest the latter one is more robust and useful.

The above insights to me is not very novel or surprise and somewhat has been addressed in exsiting literature [2]. Moreover, there is no parituclar technical novelty for devising new learning solvers.

The paper may not be comprehensive enough to make a publication in ICLR 2023.

---

### Official Review · Reviewer_F45x · 2022-10-24

**Confidence:** 4
**Correctness:** 2
**Technical Novelty And Significance:** 2
**Empirical Novelty And Significance:** 2
**Recommendation:** 3

**Clarity, Quality, Novelty And Reproducibility:**

Novelty/Quality: The impact of adding distance information has been studied in Bogyrbayeva et al https://arxiv.org/abs/2010.02369 . While this works does a more systematic study of the effect, I am not sure the presented work by itself is significant enough for ICLR. I am particularly concerned that relatively simple remedies to the problem (such as data augmentation or incorporating robustness requirements into training instance creation) were not explored, and also that authors only experiment with settings favorable to their proposed method. More systematic study of possible remedies and diverse experimental settings will strengthen the contribution of the paper.

Clarity: Main ideas of the paper was relatively easy to follow, but I had difficulties getting some details of the paper. For example, I don't think the supervision signal and the loss function for FPD model in Section 4.2 was discussed. Authors mention the goal is to imitate courier action, and cross-entropy loss could've been a straightforward choice of the loss, but these are worth mentioning.

Reproducibility: Synthetic data experiments would be easily reproduced, since authors build upon an existing open source project, and they promised to share results. Real data experiments shall not be reproduced, because they seem to be based on non-public data.

**Strength And Weaknesses:**

While the featurization of input to DL-based routing models is known to make a substantial impact to model performance, the impact of coordinate embeddings has not been systematically studied. For example, Bogyrbayeva et al https://arxiv.org/abs/2010.02369 (see Figure 6) did show that adding coordinate information makes a significant impact on model performance. Authors' systematic study provides clearer insight on the impact of coordinate information.

Another strength of the paper is that authors validate their approach on a realistic data from a large-scale production system. However, the data doesn't seem to be publicly available, which somewhat diminishes the value because others will not be able to validate the result.

My biggest concern is that it is unclear to which extent the problem raised by authors is specific to the particular experimental setting considered in this paper. Existing methods including Kool et al (which authors' implementation is based upon) does not use distance information. If authors' claim holds broadly, then they should be able to compare against Kool et al's approach on every benchmark task Kool et al considered, and demonstrate the comparative improvement. However, authors depart from the baseline paper's setting, and design a new setting that is in favor of their method: location of nodes are fixed to 10k points in training, and only at the test time they generalize to new locations. It is somewhat interesting to understand that models trained on fixed points do not generalize to new points, but if we knew the need for generalization to new points at the training time, we could've incorporated that into the creation of training instances. Similarly, if we knew the model needs rotation/translation/scaling invariance, we could've incorporated that into data augmentation step in training. Hence I am concerned authors are not showing the full picture: authors only experiment with non-standard settings which are in favor of their method. Data augmentation could've been an easy, standard remedy to the raised problem, which wasn't explored and thus significantly restricts how well authors' claim shall be generalized.

Authors seem to claim that coordinate information should be removed for routing tasks, but it is unclear to me. Distance information by itself cannot fully recover coordinate information, and thus there has to be class of routing problems which using distance information would outperform without? For example, if there are only a small number of nodes, then without the coordinate information the model will have difficulty identifying global distance structure of the graph? Clearer characterization of when one method will outperform another will be more insightful, rather than making a broad claim about the benefit of one method against another based on a particular favorable configuration.

**Summary Of The Paper:**

Authors study how the inclusion of nodes' coordinates into deep learning-based routing models affects the performance and robustness of these models. They evaluate different methods of embedding coordinates on synthetic traveling salesman problem task and FPD (Food Pickup and Delivery? Authors don't seem to spell out the acronym) task. They empirically demonstrate that only providing distance information outperforms only providing coordinate information or both.

**Summary Of The Review:**

Authors provide a good insight into the robustness of routing models to coordinates of nodes. However, the contribution is limited because the evaluation is narrowly conducted on two tasks which are also designed to be favorable to the proposed method. Also, straightforward remedies to the problem are not explored. Broader validation of the method will strengthen the contribution of the paper.

---

### Official Review · Reviewer_CEky · 2022-10-24

**Confidence:** 3
**Correctness:** 2
**Technical Novelty And Significance:** 3
**Empirical Novelty And Significance:** 3
**Recommendation:** 5

**Clarity, Quality, Novelty And Reproducibility:**

Questions:
    (1) For table 1, what does the left distribution means? Does it seem to be the city location's distribution?
    (2) You tune the hyper-parameters of the original algorithm. What if you do not tune? And is the new input format sensitive to the hyper-parameters?

**Strength And Weaknesses:**

Strength:
    (1) The observation is very simple and interesting. If this is shown to be the correct one, this can help improve the current deep models for routing problems almost with zero overhead.

Weaknesses:
    (1) Given that the claims are for all the routing problems and all architecture. So the current experiments seem to be far from enough. It would be better to consider more problems and more architecture. At least, I think one GNN-based model is essential since, as the paper mentioned, the GNN model itself also considers the distance in the edge feature.

**Summary Of The Paper:**

This paper proposes an interesting observation: when learning to solve routing problems, the distance-based input is better than the coordination based (which is currently the most used one) input.  The basic intuition behind it is that distance is an essential property of the routing problems. To validate this observation, the authors employ the current SOTA transformer models and apply them to two models (1) TSP and (2) real-world FPD. They replace the input with two types: (1) the distance-based input and (2) the concatenation of the coordination input and distance-based input.  Moreover, they design some perturbations for the test data to test the robustness of the models. The results show that the distance-based input can improve the model's performance.

**Summary Of The Review:**

This paper proposes a simple and interesting observation. However, current experiments seem not to be enough to fully support it or the authors may restrict it a little bit.

---

### Decision · Program_Chairs · 2023-01-20

**Decision:**

Reject

**Justification For Why Not Higher Score:**

The experimental validation is riddled with too many questionable aspects.

**Justification For Why Not Lower Score:**

N/A

**Metareview: Summary, Strengths And Weaknesses:**

With the expansion of online services such as logistics and food delivery, there has been renewed interest in classical routing problems such as Traveling Salesperson (TSP) and variants of Vehicle Routing. Deep Learning is emerging as an alternative to the classical OR approaches. This work claims that existing DL methods often consider the embedding of the route point coordinate as a key input, but that works using this lack robustness and generalization ability. This paper suggests abandoning the coordinates and instead use the relative distance for route point embedding. Empirical improvements are shown.

The paper’s experiments appear insufficient: more problems/architectures are needed, including at least one GNN-based model.

The paper validates its approach on data from a large-scale production system, but these data do not seem to be publicly available.

A major concern: it is unclear to what extent the problem is specific to the particular experimental setting considered in this paper. For instance, some existing methods including Kool et al. do not use distance information: if the paper’s claim holds broadly, it should be compared against Kool et al's approach on every benchmark task Kool et al. considered.
In general, the experimentation needs much more thought.